# Treatments and Outcomes in Stage I Extranodal Marginal Zone Lymphoma in the United States

**DOI:** 10.3390/cancers13081803

**Published:** 2021-04-09

**Authors:** Juan Pablo Alderuccio, Jorge A. Florindez, Isildinha M. Reis, Wei Zhao, Izidore S. Lossos

**Affiliations:** 1Department of Medicine, Division of Hematology, Sylvester Comprehensive Cancer Center, University of Miami Miller School of Medicine, Miami, FL 33136, USA; jalderuccio@med.miami.edu; 2Department of Medicine, Division of Hospital Medicine, University of Miami Miller School of Medicine, Miami, FL 33136, USA; jorgeflorindez@med.miami.edu; 3Department of Public Health Science, University of Miami Miller School of Medicine, Miami, FL 33136, USA; ireis@med.miami.edu; 4Sylvester Biostatistics and Bioinformatics Core Resource, Sylvester Comprehensive Cancer Center, University of Miami Miller School of Medicine, Miami, FL 33136, USA; wzhao2@med.miami.edu; 5Department of Molecular and Cellular Pharmacology, Sylvester Comprehensive Cancer Center, University of Miami Miller School of Medicine, Miami, FL 33136, USA

**Keywords:** extranodal marginal zone lymphoma, stage I, SEER database, treatment, survival

## Abstract

**Simple Summary:**

Extranodal marginal zone lymphoma (EMZL) is a rare disease commonly diagnosed at an early stage and remains localized for prolonged periods of time. This unique characteristic makes the use of local therapies, such as radiation therapy (RT), the preferred approach. Excellent results were previously reported implementing RT; however, majority of these studies included a small number of patients, and treatment patterns in the United States are presently unknown. Furthermore, EMZL may arise in various organs, and whether the survival is similar at different locations is unclear. In the present study, we assessed the Surveillance, Epidemiology, and End Results (SEER) database aiming to examine management and survival of localized EMZL. While differences in survival were observed by primary disease location, similar survival was observed in RT-treated stage I EMZL patients and general U.S. population matched by sex, age, and calendar year.

**Abstract:**

A considerable number of patients with extranodal marginal zone lymphoma (EMZL) are diagnosed with stage I disease. Information on treatments and survival by primary location remains limited. We extracted data from the Surveillance, Epidemiology, and End Results (SEER) database to assess treatment, primary location, and survival of patients with stage I EMZL. Results show that 7961 patients met inclusion criteria. Observation (no treatment) was the most common approach (31%) followed by radiation therapy (RT, 23%). The median overall survival (OS) was 17.3 years (95%CI 16.3 to 18.3). Shorter survival was observed in patients with stage I EMZL compared to expected survival in a cohort derived from the general U.S. population matched by sex, age, and calendar year at diagnosis. However, similar survival was observed in RT-treated patients. We identified age ≥ 60 years (SHR = 4.00, 95%CI 3.10–5.15; *p* < 0.001), higher grade transformation (SHR = 4.63, 95%CI 3.29–6.52; *p* < 0.001), and primary lung EMZL (SHR = 1.44, 95%CI 1.05–1.96; *p* = 0.022) as factors associated with shorter lymphoma-specific survival (LSS). Conversely, primary skin location (SHR = 0.50, 95%CI 0.33–0.77; *p* = 0.002) was associated with longer LSS. Our results support the use of RT as the preferred approach in localized EMZL.

## 1. Introduction

Extranodal marginal zone lymphomas (EMZLs) arise in a wide variety of mucosal sites, most commonly stomach, ocular adnexa, lung, and salivary gland [1]. EMZL generally develops in a background of chronic inflammatory disorder. The cause of chronic inflammation is site-dependent, ranging from *Helicobacter pylori* infection in gastric location to autoimmune diseases in the salivary gland (Sjögren’s syndrome) and thyroid (Hashimoto’s thyroiditis) EMZL [2]. 

The primary lymphoma location may have prognostic relevance because of organ-specific clinical manifestations, organ-specific biology, and distinct natural history [3]. However, the effect of primary EMZL location on survival remains controversial with contradictory results in the literature [4,5,6,7]. Studies from Europe (n = 401) and the United States (U.S.) (n = 497) reported longer survival and lower risk of disease relapse in gastric EMZL compared to other sites [5,6]. However, larger epidemiological studies implementing the Surveillance, Epidemiology, and End Results (SEER) database found worse outcomes in those with gastric primary location [7]. Similarly, a study using the Netherlands Cancer Registry (n = 1449) identified worse five-year overall survival (OS) in those patients with localized gastrointestinal (GI) EMZL (71%) compared to those with non-GI-EMZL (85%; *p* < 0.0001) [8]. In contrast, thyroid primary location has been consistently associated with better outcomes across studies, with a 10-year recurrence free-rate above 90% [6,9]. Nevertheless, in the majority of previous studies, the number of analyzed cases was small, thus preventing valid conclusions [10,11]. 

EMZL is diagnosed at an early stage of the disease in 60% to 80% of cases and remains localized to mucosa environment for prolonged periods of time [12,13,14]. This unique biological characteristic makes local treatment, such as eradication of *H. pylori* infection from the stomach or radiation therapy (RT) in other locations, the preferred approach. The majority of gastric EMZLs are associated with chronic *H. pylori* infection, and its eradication leads to long-term disease-free survival in 60% of the patients [15,16,17]. Successful *H. pylori* eradication strategies in northern Italy have been associated with a decreased incidence of gastric EMZL in that area [18]. Current guidelines recommend implementation of RT in patients with *H. pylori*-associated gastric EMZL not achieving remission after antibiotic therapy or stage I EMZL at other primary locations. RT was repeatedly shown to be a highly effective treatment [1,6,19,20]. However, some physicians are reluctant to use RT in view of potential side effects and despite improved safety of modern techniques [21]. Data in follicular lymphoma (FL) demonstrated underuse of RT in stage I disease despite proven efficacy [22]. Also in EMZL, approaches used in practice are variable with many patients managed with either observation or chemotherapy despite staging workup indicating localized disease [6,21,23]. 

In this study, we conducted a SEER-based population analysis focusing on stage I EMZL. We aimed to describe outcomes by disease primary location and specifically with RT. We also compared survival of SEER patient cohort to expected survival in a cohort derived from the general U.S. population matched by sex, age, and calendar year at diagnosis. 

## 2. Methods

The SEER database of the National Cancer Institute (NCI) in the United States was used to identify patients with EMZL (International Classification of Diseases for Oncology (ICD-O-3) codes 9689/3) [24]. The study data were acquired from the April 2019 release of the SEER database using the SEER*STAT software version 8.3.6, selecting EMZL subjects diagnosed between 1995 and 2015 with follow-up through 2016 from 18 SEER registries encompassing the following states: California, Georgia, Connecticut, Illinois, Hawaii, Iowa, New Mexico, Washington, Utah, Kentucky, and Louisiana. Patients with unknown survival data, recorded survival time of zero (0) days, without histologic confirmation of EMZL, unknown cause of death, unknown primary site, non-stage I, age < 18 years, and EMZL not listed as a first primary malignancy were excluded from the analysis (Figure 1). Staging modality, indications for therapy, tumor recurrences, or treatments used at the time of progression are not recorded. A decision to pursue watchful waiting and refusal of treatment or antibiotic therapy is recorded as no initial therapy. Available data do not contain information about *H. pylori* eradication, RT dose, field, and fraction schedule. There is also no information on response to treatment, toxicities, or duration of remission. In the SEER database, rituximab is considered chemotherapy for diagnosis years 1999 to 2012, and it is excluded from chemotherapy variable after 2012. The treatment given is capturing treatment not only at diagnosis but throughout the course of EMZL. When chemotherapy is identified as “yes”, SEER does not separate rituximab from conventional chemotherapy or the number of therapy lines. This study was conducted according to the guidelines of the Declaration of Helsinki, and because SEER data are publicly available, no approval was requested from the University of Miami IRB or Ethical Committee.

### Statistical Analyses

Descriptive statistics were used to summarize demographic, tumor, and treatment characteristics. Treatment included chemotherapy, RT or surgery only, and combinations of two (chemotherapy and RT, chemotherapy and surgery, surgery and RT) or three treatment modalities (surgery, radiation, and chemotherapy). OS and lymphoma-specific survival (LSS) were the primary endpoints of this study. Subset analyses were performed for RT only and RT and surgery subgroups. OS was estimated by the Kaplan–Meier method; confidence intervals (CIs) for survival rate at specific times were calculated based on the log-log transformation method. The logrank test was used for comparison of survival curves. The Cox proportional hazards regression model was used to assess predictors of OS. Unadjusted and adjusted hazard ratios (HRs) and corresponding 95% CIs and *p*-values were reported. Competing risk analysis was used to evaluate LSS. The cumulative incidence method was used to estimate cumulative incidence rate (CIR) of lymphoma-specific death (LSD); cumulative incidence curves were compared using Gray’s test, and the Fine and Gray subdistribution hazard model was used to assess the effect of demographic, tumor, and treatment-related variables on the risk of LSD. Unadjusted and adjusted subdistribution hazard ratios (SHRs) with corresponding 95% CIs and *p*-values are reported. 

We also compared survival of this SEER patient cohort to expected survival in a cohort of the general U.S. population matched by sex, age, and calendar year at diagnosis. The expected survival curve in the matched U.S. population cohort was generated in R using the survexp.us population in the “survival” R package and the Ederer approach, which assumes complete follow-up of 20 years [25,26]. We used the one-sample logrank test to test whether survival in the SEER stage I EMZL cohort is similar to expected survival in the matched U.S. population cohort. The one-sample logrank test is a test of whether the standardized mortality ratio (SMR) is equal to 1 (similar mortality), that is, equivalent to test whether survival in the SEER cohort is similar to expected survival in a matched cohort. Thus, *p* > 0.05 indicates equality of mortality, or equivalently, similar survival in both cohorts. The standardized mortality ratio (SMR) is the ratio of the number of observed deaths to expected number of deaths in the study population under the assumption that the mortality rates for the study population are the same as those for the matched population [27,28]. 

## 3. Results

A total of 7961 patients with stage I EMZL were included in this analysis. The median age at diagnosis was 64 years (range 18 to 101), and 60% ≥ 60 years old. EMZL was slightly more common in women (55.5%), and patients were mainly of non-Hispanic origin (88%). EMZL primary locations were as follows: gastric (38.1%), skin (13.2%), ocular adnexa (12.4%), GI non-gastric (8.6%), salivary gland (8.1%), lung (7.9%), breast (3.3%), oral cavity (2.6%), thyroid (2.3%), and others (3.6%). Higher grade transformation (HGT) was a rare event occurring in only 1.2% of the patients. The following treatments were used: RT (23%), surgery (19.2%), and chemotherapy (8%) as single modalities, combined modalities (18.5%), and observation (31.3%) (Table 1).

### 3.1. Overall Survival

There were 2073 (26%) deaths, and 5888 (74%) patients remained alive at last follow-up. The estimated number of expected deaths (E) would be 1749, based on the death rates of the cohort of the general U.S. population matched by sex, age, and calendar year of diagnosis. Thus, the estimated SMR was 1.19 (95%CI 1.14 to 1.24), *p* < 0.001, indicating significantly increased mortality in the SEER cohort of stage I EMZL patients compared to that expected in the matched cohort. The median OS in the whole SEER stage I EMZL cohort was 17.3 years (95%CI 16.3–18.3 years), and the 5- and 10-year OS rates were 85% (95%CI 84.1–85.8%) and 70.2% (95%CI 68.9–71.5%), respectively. In the matched cohort of the general U.S. population, the expected five-year and 10-year OS rates would be 86.5% and 73%, respectively (Figure 2A). 

In univariable analyses (Table 2), OS was longer in patients with skin (10-year OS 82.3%, 95%CI 78.7–85.4%; HR = 0.39, 95%CI 0.32–0.47, *p* < 0.001), thyroid (10-year OS 78.7%, 95%CI 70.0–85.1%; HR = 0.52, 95%CI 0.37–0.74, *p* < 0.001), ocular adnexa (10-year OS 75.4%, 95%CI 72.2–78.3%; HR = 0.65, 95%CI 0.57–0.74, *p* < 0.001), GI non-gastric (10-year OS 71.3%, 95%CI 66.7–75.3%; HR = 0.77, 95%CI 0.65–0.90, *p* = 0.001), salivary gland (10-year OS 74.6%, 95%CI 70.1–78.5%, HR = 0.69, 95%CI 0.58–0.81, *p* < 0.001), and oral (10-year OS 70.6%, 95%CI 62.2–77.5%; HR = 0.77, 95%CI 0.59–1.00, *p* = 0.05) primary locations, compared to patients with gastric primary location (10-year OS 64.5%, 95%CI 62.4–66.6%) as the reference group. Survival in patients with lung (10-year OS 62.7%, 95%CI 57.3–67.7%; HR = 0.90, 95%CI 0.76–1.06, *p* = 0.209) and breast (10-year OS 68.5%, 95%CI 60.5–75.2%; HR = 0.87, 95%CI 0.69–1.11, *p* = 0.280) primary locations were similar to that of patients with gastric primary location (Figure 2B). The cohort of combined treatment modalities (10-year OS 78.7%, 95%CI 76.1–81.1%; HR = 0.91, 95%CI 0.79–1.06, *p* = 0.244) had similar survival to single therapy with RT (10-year OS 75.7%, 95%CI 72.9–78.3%;), while surgery only (10-year OS 70%, 95%CI 67.0–72.8%, HR = 1.25, 95%CI 1.08–1.44, *p* = 0.003), chemotherapy only (10-year OS 64%, 95%CI 59.2–68.4%, HR = 1.67, 95%CI 1.41–1.99, *p* < 0.001), and observation (10-year OS 62.6%, 95%CI 60.2–64.9%, HR = 1.70, 95%CI 1.50–1.93, *p* < 0.001) had worse survival than RT only (Figure 2C, Table 2). We next analyzed outcomes of individual combination therapies versus RT alone (Figure 2D, Table 2). Combination of surgery and RT (10-year OS 82.1%, 95%CI 78.8–84.9%) and surgery, chemotherapy, and RT (10-year OS 81.0%, 95%CI 69.7–88.4%) were associated with best survivals (Figure 2D). In univariable analysis, the combination of surgery and RT was statistically significantly better than RT alone (HR = 0.77, 95%CI 0.64–0.93, *p* = 0.005), but the combination of surgery, chemotherapy, and RT was not statistically significantly different from RT alone (HR = 0.69, 95%CI 0.43–1.11, *p* = 0.129), most likely due to the small number of patients in that group (n = 94). There was no statistically significant difference between “surgery and RT” versus “surgery, chemotherapy, and RT” combined modalities, suggesting that addition of chemotherapy to surgery and RT is not improving survival (*p* = 0.665). Similarly, adding chemotherapy to RT (10-year OS 73.2%, 95%CI 66.1–79.0%; HR = 1.24, 95%CI 0.95–1.62, *p* = 0.117) did not improve survival compared to RT only (10-year OS 75.7%, 95%CI 72.9–78.3%). 

We next performed multivariable analyses. Multivariable Cox model included adjustment for five treatment modalities as a result of grouping the combined modalities together. This analysis confirmed better survival for primary skin (HR = 0.59, 95%CI 0.48–0.71, *p* < 0.001), thyroid (HR = 0.68, 95%CI 0.47–0.97, *p* = 0.032), and GI non-gastric (HR = 0.82, 95%CI 0.69–0.98, *p* = 0.025) locations compared to gastric primary location (Table 2). Multivariable Cox model identified significantly shorter OS for chemotherapy only (HR = 1.46, 95%CI 1.23–1.74, *p* < 0.001), surgery only (HR = 1.28, 95%CI 1.09–1.50, *p* = 0.002), and observation (HR = 1.47, 95%CI 1.29–1.67, *p* < 0.001), compared to radiation only as the reference. Combined treatment modalities as a group was not significantly different from RT only (HR = 1.0, *p* = 0.970) (Figure 2C, Table 2). 

Since patients treated with RT achieved longer survival than other individual modalities, we next compared their survival to general U.S. population matched by sex, age, and calendar year of diagnosis. Notably, patients treated with RT had similar survival compared to matched U.S. population cohort with 5- and 10-year OS rates of 89.2% (95%CI 87.5–78.3%), and 75.7% (95%CI 72.9–78.3%) vs. 88.1% and 75.4% in the matched cohort, respectively. The estimated SMR was 1.01 (95%CI 0.91–1.12), *p* = 0.799, comparing 347 deaths in the SEER cohort versus 342.3 expected deaths in the matched cohort (Figure 3A). In patients treated with RT only, skin primary location (10-year OS 87.1%, 95%CI 79.3–92.1%; HR = 0.34, 95%CI 0.21–0.54, *p* < 0.001) had significant better OS than gastric location (10-year OS 72.4%, 95%CI 67.9–76.4%), and OS in lung presentation (10-year OS 58.0%, 95%CI 31.6–77.3%; HR = 0.87, 95%CI 0.46–1.65, *p* = 0.673) was not statistically significantly different from OS in gastric primary location. Similar results were observed in multivariable analysis (*p* < 0.001 for skin and *p* = 0.215 for lung versus gastric primary location) (Figure 3B, Table 3). 

### 3.2. Overall Survival by Treatment Strategy within EMZL Location Subgroups 

We analyzed the effect of treatment approach on OS in subgroups of primary locations with a large number of patients and events. In gastric EMZL, RT with surgery and/or chemotherapy showed no difference in OS in comparison to RT only (10-year OS 77.0%, 95%CI 68.9–83.3% versus 72.4%, 95%CI 67.9–76.4%, respectively; HR = 1.04, *p* = 0.826), while observation (10-year OS 59.9%; HR = 1.69, *p* < 0.0001), surgery only (10-year OS 67.3%; HR = 1.41, *p* < 0.038), and chemotherapy only (10-year OS 60.2%; HR = 1.66, *p* < 0.0001) were associated with shorter OS than RT only (Appendix A). In skin EMZL, RT only and RT with surgery/chemotherapy displayed similar survival (10-year OS 87.1%, 95%CI 79.3–92.1%, and 87.4%, 95%CI 80.1–92.1%, respectively; HR 1.34, *p* = 0.327). Conversely, shorter survival was observed in chemotherapy only (10-year OS 65.8%, 95%CI 37.3–83.7%) and observation (10-year OS 76.6%, 95%CI 67–83.7%), and 10-year OS in these groups were significantly lower than in RT alone (HR = 3.00, *p* = 0.0067, and HR = 2.21, *p* = 0.0066, respectively) (Appendix A). In primary ocular adnexa location, there was no difference in OS survival between RT only and the other treatments, with the exception of the combination of surgery and chemotherapy (10-year OS 38.1%, 95%CI 20.1–56% versus 78.6% 95%CI 73.2–83.1% in RT only; HR = 3.45, 95%CI 2.08–5.74, *p* < 0.0001) (Appendix A), but the number of patients in the latter group was small. In salivary gland primary location, there were no significant differences in OS across all treatment approaches (Appendix A). In primary lung location, patients on observation demonstrated significant inferior OS compared to RT only (*p* = 0.0176) (Appendix A). 

### 3.3. Lymphoma-Specific Survival (LSS)

A total of 515 (6.5%) EMZL-related deaths occurred, whereas 1558 (19.6%) died of non-EMZL causes. The cumulative incidence rate (CIR) of lymphoma-specific death at 10 years was 7.6% (6.9–8.3%) (Figure 3C). Compared to gastric EMZL (reference; 10-year CIR 8.2%, 95%CI 7.1–9.4%), lung location exhibited higher risk of death (10-year CIR 13.4%, 95%CI 10.2–17.1%; SHR = 1.48, 95CI% 1.11–1.97, *p* = 0.007), while skin (10-year CIR 3.7%, 95%CI 2.3–5.6%; SHR = 0.36, 95%CI 0.24–0.55, *p* < 0.001), and ocular adnexa (10-year CIR 4.7%, 95%CI 3.4–6.3%; SHR = 0.60, 95%CI 0.44–0.82, *p* = 0.001) showed lower risk of lymphoma-specific death. There was no significant difference between thyroid (10-year CIR 4.6%, 95%CI 2.0–8.8%; SHR = 0.62, 95%CI 0.31–126, *p* = 0.186) and gastric primary location (Figure 3D and Table 4). All treatment modalities, with exception of surgery and chemotherapy, showed statistically significantly better LSS compared to chemotherapy only (HRs between 0.31 and 0.52, *p* < 0.05); in particular, the risk of death from lymphoma was lowest in patients that underwent a combined RT with surgery modality (SHR = 0.31, 95%CI 0.21–0.45; *p* < 0.001) (data not shown). Multivariable competing-risk analysis demonstrated the following characteristics associated with shorter LSS: age ≥ 60 years (SHR = 4.00, 95%CI 3.10–5.15, *p* < 0.001), HGT (SHR = 4.63, 95%CI 3.29–6.52, *p* < 0.001), lung EMZL (SHR = 1.44, 95%CI 1.05–1.96, *p* = 0.022), and other primary locations (SHR = 1.59, 95%CI 1.05–2.40, *p* = 0.029). Conversely, the primary skin location (SHR = 0.50, 95%CI 0.33–0.77, *p* = 0.002) was associated with longer LSS. Of note, in the multivariable model the difference between ocular adnexa and gastric primary location was no longer statistically significant (HR = 0.75, 95%CI 0.54–1.04, *p* = 0.086), and chemotherapy use was associated with shorter LSS compared to RT (SHR = 2.30, 95%CI 1.68–3.15, *p* < 0.001). Same variables were also significant in multivariable model 2. When analyzing single and combined approaches, chemotherapy only was associated with worse survival (SHR = 2.33, 95%CI 1.70–3.19, *p* < 0.001) followed by surgery plus chemotherapy (SHR= 1.96, 95%CI 1.23–3.11, *p* = 0.005) (Table 4). In patients treated with RT only (n = 1835), age ≥60 years (SHR = 5.69, 95%CI 2.80–11.6, *p* < 0.001) and HGT (SHR = 4.92, 95%CI 2.01–12.0, *p* < 0.001) were the only variables associated with shorter LSS at the 5% significance level on the basis of a multivariable model. Notably, EMZL location did not affect LSS indicating that RT is equally effective irrespective of primary location (Table 3). Same prognostic variables were identified in patients treated with RT and surgery (Appendix A). Non-lymphoma-related causes of death are shown in Appendix A. 

## 4. Discussion

This study represents the largest analysis evaluating prognostic factors, treatment strategies, and treatment-related outcomes in stage I EMZL. Patients with EMZL experience excellent survival with a 10-year OS of 70.2%. However, we observed previously not reported shorter OS compared to matched-U.S. population irrespective of treatment modalities. This difference became not significant when RT was implemented, confirming this approach as the preferred treatment in stage I EMZL. Similar to prior reports in FL [22], we observed that RT was used in only 23% of patients indicating underuse of the most effective therapy. A previous study in FL demonstrated longer progression-free survival combining RT and chemotherapy [29]. We did not observe better LSS and OS survival in EMZL patients treated with this combination.

RT has been associated with excellent disease control across studies in EMZL [6,9,30]. However, we found its underutilization in our analysis. Similar findings were described by Ling et al. analyzing 22,378 patients with MZL included in the National Cancer Database. Authors found a decrease in rates of RT use over the years from a peak in 2007 (39%) to a significantly lower rate in 2011 (33%; *p* < 0.001) that correlated with an increase in the implementation of systemic therapy in MZL. On multivariable propensity score-adjusted survival analysis, RT remained independently associated with better OS (hazard of death, 0.75, 95%CI 0.65–0.85; *p* < 0.001) [21]. Moreover, Olszewski et al. identified factors associated with lower use of RT in patients with stage I/II EMZL including age older than 70 years (OR = 0.70, 95%CI 0.59–0.83; *p* = 0.0001), Hispanic (OR = 0.67, 95%CI 0.50–0.89; *p* = 0.006), Black ethnicity (OR = 0.48, 95%CI 0.33–0.69; *p* = 0.0001), non-Asians women (OR = 0.83, 95%CI 0.73–0.94; *p* = 0.005), areas with higher poverty levels (OR = 0.71, 95%CI 0.60–0.85; *p* = 0.0002), rural counties (OR = 0.52, 95%CI 0.31–0.87; *p* = 0.01), uninsured patients (OR = 0.45, 95%CI 0.22–0.92; *p* = 0.03), and state-sponsored Medicaid insurance (OR = 0.55, 95%CI 0.38–0.80; *p* = 0.002) [23]. EMZL-related clinical variables were not among the factors driving the decision not to use RT. Similar findings were reported in the National Cancer Base Analysis on early-stage FL. In that study, the following factors were associated with decreased use of RT: increasing age, female sex, African American race, increase comorbidity score, treatment at an academic/research program, stage II disease, presence of B symptoms, and more recent years of diagnosis [31]. 

In our study, RT was associated with excellent outcomes in all primary locations with a 10-year OS of 78.7% and lower risk of lymphoma-related death (SHR = 0.31). We identified age ≥ 60 years (HR = 4.00) and HGT (HR = 4.63) as risk factors for shorter LSS. Importantly, we did not observe significant survival differences in patients treated with an RT-based approach across all primary locations. Based on these results RT-based therapy should remain the frontline therapy in stage I EMZL, with exception of gastric EMZL with *H. pylori* infection, which should be first treated with antibiotics. In gastric EMZL, RT should be used only in patients who failed this treatment or in whom lymphoma is not associated with this infection. We acknowledge that RT may not be feasible in some anatomic locations such as GI-nongastric, liver, kidney, or salivary gland, where long-term toxicity may outweigh benefits. Further, previous reports on different outcomes in patients with stage I EMZL might result from differences in therapeutic approaches across the studies. Paucity of data exists regarding long-term survival in patients with localized EMZL compared to general population. A small retrospective study (n = 49) in Norway found similar median life expectancy for patients with EMZL compared to sex- and age-matched controls (79 vs. 83.6 years) [17]. Contrary to this observation, our analysis demonstrated for the first-time shorter survival in patients with stage I EMZL compared to general U.S. population which can be overcome by the implementation of RT as initial therapy. EMZL harbors a persistent risk of disease relapse requiring long-term follow-up. Our group previously demonstrated a continuous risk of lymphoma relapse with an estimated cumulative relapse of 31% at 10 years in patients with primary ocular adnexal EMZL [30]. Similarly, Raderer et al. reported a relapse rate of 37% with a median time of relapse of 47 months (range 14 to 307 months) after achieving initial complete response in EMZL [14]. However, the presence of effective second-line therapies in these patients may decrease patients’ mortality resulting in similar survival.

The clinical course of patients with EMZL is characterized by an increased risk of HGT, which is an independent risk factor for poor outcomes. HGT is a rare event occurring in 4% to 8% of EMZL [32,33,34]. The incidence of HGT in this analysis was only 1.2%, however, associated with shorter OS and LSS in all multivariable models. Moreover, HGT was the most significant risk factor for shorter LSS. The true incidence of HGT in stage I EMZL remains unclear, but in a large study analyzing transformation, this event occurred in 7% of patients with a limited stage. In this study, the risk of HGT by 15 years was 4% compared to 16% in patients with advanced-stage disease [34]. These results resemble FL data where advanced-stage disease is associated with a higher risk of HGT; however, the incidence of transformation in EMZL is markedly lower [35,36]. 

Thyroid and gastric EMZL treated with RT have been associated with a lower risk of disease relapse and long-term survival. Tsang et al. did not observe relapses in these locations after a median follow-up of 5.1 years [37]. Similarly, the International Extranodal Lymphoma Study Group reported a 10-year freedom from treatment failure and OS of 88% and 70%, respectively, in patients with gastric EMZL [38]. We observed better survival in skin and thyroid primary locations; however, gastric primary location was associated with shorter survival, contradicting prior studies [5,10]. Factors influencing geographical differences are unclear, but we hypothesize that underuse of RT and *H. pylori* infection incidence are the explanation of such discrepancies. A population-based study carried out in the north of Italy demonstrated decreasing incidence in *H. pylori*-related gastric EMZL from 65% in the 1997–2001 period to 19% in the 2002–2007 period (*p* = 0.007) [39]. This decrease in association with *H. pylori* infection may result in different disease biology of gastric EMZL. In our study, 3031 patients with gastric EMZL were included. Of those patients, no therapy was recorded in the SEER database for 1543 (50.9%) patients. SEER database does not record the presence of *H. pylori infection* and *H. pylori* antibiotic therapy, but a significant proportion of these patients might be classified under the observation group. However, the exact number of these patients cannot be estimated from the SEER database. We also cannot assume that all these patients were treated with antibiotics, since survival of gastric EMZL stage I patients in the observation group was inferior compared to patients treated with RT. Absence of this data represents one of the major caveats in presented analysis. If this data would be available, we might better understand the findings of inferior outcome of gastric EMZL in the SEER database. 

Further, the SEER database also does not collect data on diagnostic procedures used to establish the lymphoma stage. Extensive staging work-up was postulated by Raderer et al. in EMZL. The authors recommended that staging should include ophthalmologic examination, ear, nose, and throat investigation including sonography or magnetic resonance imaging of the salivary glands and lacrimal glands, endoscopies with multiple biopsies of the GI tract, computed tomography of thorax and abdomen, sonography of cervical, inguinal, and axillary lymph nodes, and bone marrow biopsy. By applying this approach, 25% of their patients with gastric EMZL had multiorgan involvement beyond the GI tract, and 46% of patients with extragastric EMZL also presented with advanced disease. Survival was not affected by extension of disease and this approach has not been generally accepted [40]. Furthermore, our group demonstrated the absence of prognostic value of staging bone marrow biopsy in patients with clinical/radiological localized EMZL treated with frontline RT [41]. In this study, staging bone marrow biopsy did not affect lymphoma-specific survival providing further rationale against extensive work up in EMZL.

The present study carries limitations of large epidemiological studies using the SEER database such as lack of information on staging modality, indications for therapy, use of antibiotic therapy in patients with gastric EMZL, *H. pylori* eradication, relapses, progression-free survival, radiation doses, chemotherapy regimens, and treatment-related adverse events. Clinical and laboratory variables are not available in the SEER database preventing the confirmation of previously described predictors of increased risk of transformation. Despite these limitations, the SEER database allows statistically powered analysis of a large number of patients with a rare and heterogeneous disease such as EMZL. 

## 5. Conclusions

In the absence of randomized studies, this analysis supports the use of RT as the preferred approach in stage I EMZL. We do not have information on RT doses and approaches used in this cohort. Implementing the smallest required dose that can produce persistent local control is needed, but what this dose is remains unknown. Randomized trials of different RT doses are needed in EMZL patients to address this important question. In addition, studies are needed to discover novel, potentially less toxic therapies that will improve survival of these patients. Further studies addressing EMZL biology at different geographical locations are needed to elucidate survival characteristics at specific primary locations (e.g., gastric) observed in studies originating in U.S. and non-U.S. populations.

## Figures and Tables

**Figure 1 cancers-13-01803-f001:**
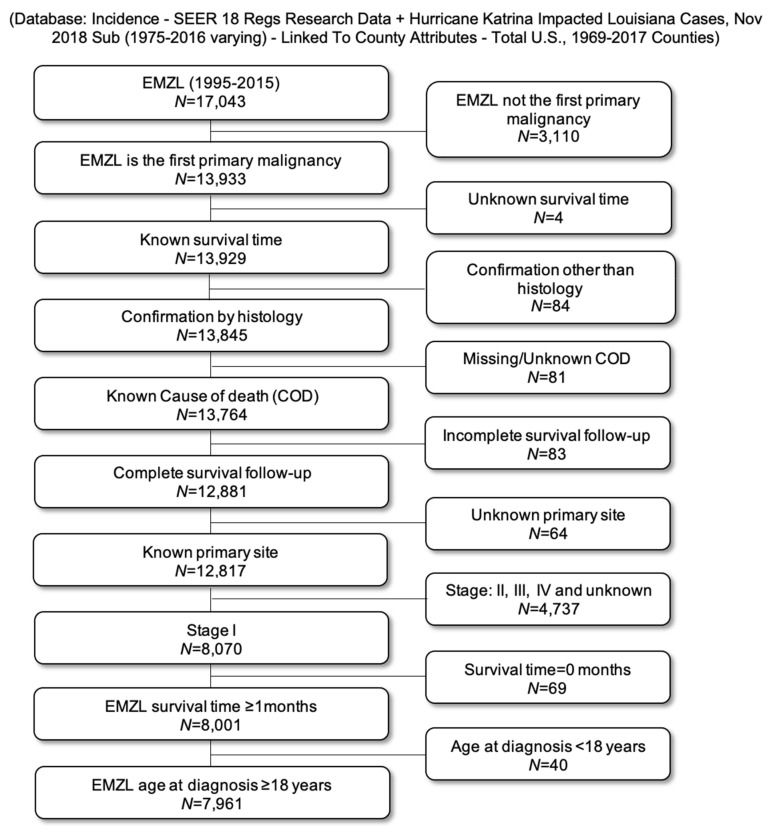
Consort chart of patient selection.

**Figure 2 cancers-13-01803-f002:**
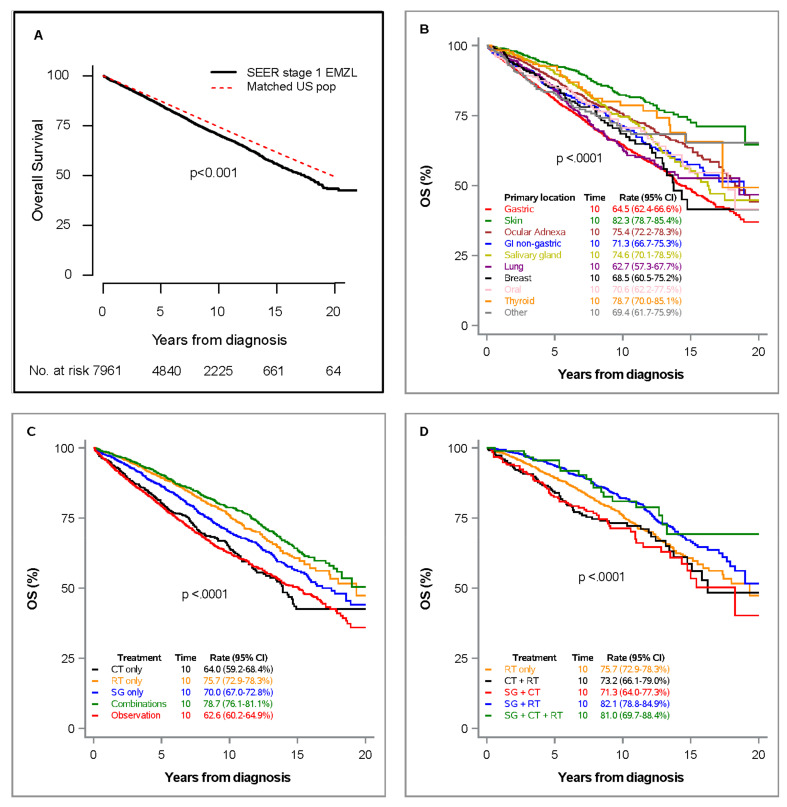
Overall survival in SEER stage I EMZL cohort vs. age-, sex-, and calendar-matched U.S. population cohort (**A**); overall survival in SEER stage I EMZL cohort by primary location (**B**), by treatment modality (**C**), and by combined treatment modality and RT only (**D**). Tick marks for censored observations not shown given the large number of patients.

**Figure 3 cancers-13-01803-f003:**
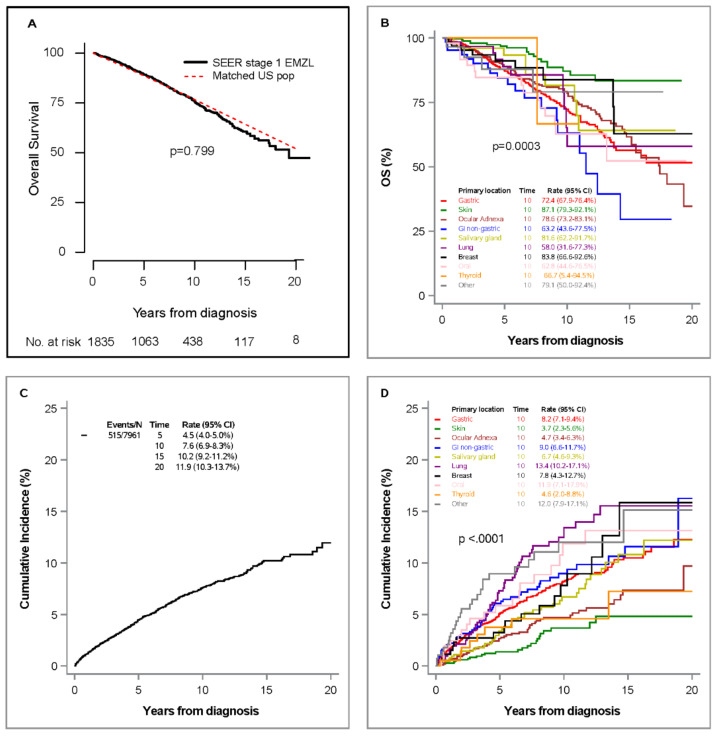
Overall survival in patients with stage I extranodal marginal zone lymphoma treated with radiation only (n = 1835) versus age-, sex-, and calendar-matched U.S. population cohort (**A**); overall survival in patients with stage I EMZL treated with radiation only (n = 1835) by primary location (**B**); and cumulative incidence rate of lymphoma-specific death in stage I EMZL overall (**C**) and by primary location (**D**). Tick marks for censored observations not shown given the large number of patients.

**Table 1 cancers-13-01803-t001:** Characteristics of stage I EMZL cohort.

Variable	N	%
**Total**	**7961**	**100.0**
**Age**		
<60 years	3184	40.0
≥60 years	4777	60.0
**Sex**		
Female	4422	55.5
Male	3539	44.5
**Race**		
White	6284	78.9
Black	685	8.6
Other	992	12.5
**Hispanic**		
Non-Hispanic	7002	88.0
Hispanic	959	12.0
**Primary location**		
Gastric	3031	38.1
Skin	1051	13.2
Ocular adnexa	990	12.4
GI non-gastric	682	8.6
Salivary gland	644	8.1
Lung	627	7.9
Breast	262	3.3
Oral	206	2.6
Thyroid	183	2.3
Other	285	3.6
**DLBCL transformation**		
No	7869	98.8
Yes	92	1.2
**Chemotherapy**		
No/Unknown *	6752	84.8
Yes	1209	15.2
**Radiotherapy**		
No/Unknown *	4908	61.7
Yes	3053	38.3
**Surgery**		
No	5100	64.1
Yes	2775	34.9
Unknown	86	1.1
**Treatment**		
Radiation (RT) only	1835	23.0
Surgery (SG) only	1528	19.2
Chemotherapy (CT) only	640	8.0
Combinations	1470	18.5
SG + RT	901	11.3
SG + CT	252	3.2
CT + RT	223	2.8
SG + CT + RT	94	1.2
Observation	2488	31.3
**Vital status**		
Alive	5888	74.0
Death	2073	26.0
Death from lymphoma	515	6.5
Death from other cause	1558	19.6

* Chemotherapy and radiotherapy may include cases of “unknown”. Surveillance, Epidemiology, and End Results (SEER) classifies chemotherapy data as “yes—patient had chemotherapy” and “no/unknown—no evidence of chemotherapy was found in the medical records examined”. Abbreviations: EMZL, extranodal marginal zone lymphoma; GI, gastrointestinal; DLBCL, diffuse large B cell lymphoma.

**Table 2 cancers-13-01803-t002:** Cox models for overall survival in stage I EMZL cohort (N = 7961, including 2073 death events).

		Univariable		Multivariable Model	
Variable	Category	HR (95% CI)	*p*	HR (95% CI)	*p*
**Age**	<60 years	Reference		Reference	
	≥60 years	6.69 (5.85, 7.65)	<0.001	6.29 (5.49, 7.20)	<0.001
**Sex**	Female	Reference		Reference	
	Male	1.17 (1.08, 1.28)	<0.001	1.25 (1.14, 1.36)	<0.001
**Race**	Non-Hispanic white	Reference		Reference	
	Black	0.94 (0.80, 1.09)	0.405	1.04 (0.89, 1.22)	0.620
	Other	0.70 (0.61, 0.81)	<0.001	0.75 (0.65, 0.87)	<0.001
**Ethnicity**	Non-Hispanic	Reference		Reference	
	Hispanic	0.83 (0.72, 0.96)	0.013	0.93 (0.80, 1.07)	0.299
**Primary location**	Gastric	Reference		Reference	
	Skin	0.39 (0.32, 0.47)	<0.001	0.59 (0.48, 0.71)	<0.001
	Ocular adnexa	0.65 (0.57, 0.74)	<0.001	0.90 (0.78, 1.05)	0.172
	GI non-gastric	0.77 (0.65, 0.90)	0.001	0.82 (0.69, 0.98)	0.025
	Salivary gland	0.69 (0.58, 0.81)	<0.001	0.92 (0.76, 1.10)	0.354
	Lung	0.90 (0.76, 1.06)	0.209	0.87 (0.73, 1.04)	0.136
	Breast	0.87 (0.69, 1.11)	0.280	1.04 (0.81, 1.34)	0.752
	Oral	0.77 (0.59, 1.00)	0.050	0.94 (0.71, 1.23)	0.630
	Thyroid	0.52 (0.37, 0.74)	<0.001	0.68 (0.47, 0.97)	0.032
	Other	0.74 (0.57, 0.96)	0.025	0.86 (0.66, 1.12)	0.258
**DLBCL transformation**	No	Reference		Reference	
	Yes	1.67 (1.24, 2.26)	<0.001	1.49 (1.10, 2.02)	0.009
**Treatment**	RT only	Reference		Reference	
	CT only	1.67 (1.41, 1.99)	<0.001	1.46 (1.23, 1.74)	<0.001
	SG only	1.25 (1.08, 1.44)	0.003	1.28 (1.09, 1.50)	0.002
	Combinations	0.91 (0.79, 1.06)	0.244	1.00 (0.85, 1.17)	0.970
	Observation	1.70 (1.50, 1.93)	<0.001	1.47 (1.29, 1.67)	<0.001
**Treatment**	RT only	Reference		Not included	
	CT only	1.67 (1.41, 1.99)	<0.001		
	SG only	1.25 (1.08, 1.44)	0.003		
	CT + RT	1.24 (0.95, 1.62)	0.117		
	SG + CT	1.32 (1.02, 1.71)	0.037		
	SG + RT	0.77 (0.64, 0.93)	0.005		
	SG + CT + RT	0.69 (0.43, 1.11)	0.129		
	Observation	1.70 (1.50, 1.93)	<0.001		

HR: hazard ratio. 95% CI: 95% confidence interval. Abbreviations: EMZL, extranodal marginal zone lymphoma; GI, gastrointestinal; DLBCL, diffuse large B cell lymphoma; CT, chemotherapy; RT, radiation therapy; SG, surgery.

**Table 3 cancers-13-01803-t003:** Univariable and multivariable analyses for OS and LSS in stage I EMZL that was treated with radiation only (N = 1835).

		Overall Survival (OS)smakal(347 Deaths)				Lymphoma-Specific Survival (LSS)smakal(78 Deaths from Lymphoma and 269 Deaths from Other Cause as Competing Risk)			
		Univariable		Multivariable		Univariable		Multivariable	
Variable	Category	HR (95%CI)	*p*	HR (95%CI)	*p*	SHR (95%CI)	*p*	SHR (95%CI)	*p*
**Age**	<60	Reference		Reference		Reference		Reference	
	≥60	7.83 (5.66, 10.8)	<0.001	7.58 (5.46, 10.5)	<0.001	6.31 (3.17, 12.6)	<0.001	5.69 (2.80, 11.6)	<0.001
**Sex**	Female	Reference		Reference		Reference		Reference	
	Male	1.13 (0.92, 1.40)	0.242	1.24 (0.99, 1.54)	0.058	0.95 (0.61, 1.48)	0.813	1.06 (0.66, 1.71)	0.803
**Race**	Non-Hispanic White	Reference		Reference		Reference		Reference	
	Black	1.06 (0.72, 1.56)	0.773	1.20 (0.81, 1.79)	0.361	0.86 (0.34, 2.16)	0.748	0.99 (0.39, 2.48)	0.984
	Other	0.87 (0.63, 1.20)	0.395	0.97 (0.70, 1.34)	0.853	1.34 (0.75, 2.38)	0.327	1.34 (0.71, 2.53)	0.363
**Ethnicity**	Non-Hispanic	Reference		Reference		Reference		Reference	
	Hispanic	0.68 (0.43, 1.07)	0.097	0.85 (0.53, 1.36)	0.494	0.47 (0.15, 1.49)	0.200	0.50 (0.14, 1.78)	0.285
**Primary location**	Gastric	Reference		Reference		Reference		Reference	
	Skin	0.34 (0.21, 0.54)	<0.001	0.44 (0.27, 0.71)	<0.001	0.44 (0.17, 1.13)	0.089	0.58 (0.22, 1.50)	0.262
	Ocular adnexa	0.88 (0.68, 1.14)	0.340	1.00 (0.77, 1.30)	0.980	0.79 (0.44, 1.42)	0.431	0.84 (0.47, 1.52)	0.572
	GI non-gastric	1.54 (0.96, 2.47)	0.075	1.46 (0.91, 2.36)	0.116	1.74 (0.69, 4.42)	0.244	1.48 (0.56, 3.87)	0.427
	Salivary gland	0.77 (0.39, 1.50)	0.440	1.04 (0.53, 2.04)	0.902	1.28 (0.40, 4.14)	0.680	1.70 (0.53, 5.48)	0.371
	Lung	0.87 (0.46, 1.65)	0.673	0.67 (0.35, 1.27)	0.215	1.61 (0.57, 4.52)	0.367	1.36 (0.48, 3.90)	0.561
	Breast	0.67 (0.34, 1.30)	0.236	0.76 (0.39, 1.51)	0.437	0.35 (0.05, 2.64)	0.311	0.36 (0.05, 2.78)	0.325
	Oral	1.24 (0.74, 2.07)	0.413	1.34 (0.78, 2.31)	0.283	1.86 (0.73, 4.76)	0.194	1.28 (0.41, 3.99)	0.665
	Thyroid	0.68 (0.10, 4.86)	0.701	0.99 (0.14, 7.10)	0.988	NE		NE	
	Other	0.75 (0.28, 2.01)	0.563	0.79 (0.29, 2.17)	0.652	1.77 (0.43, 7.37)	0.431	1.32 (0.27, 6.51)	0.736
**DLBCL transformation**	No	Reference		Reference		Reference		Reference	
	Yes	1.94 (1.09, 3.45)	0.025	1.35 (0.73, 2.52)	0.340	6.79 (3.28, 14.1)	<0.001	4.92 (2.01, 12.0)	<0.001

HR: hazard ratio from Cox models. SHR: subdistribution hazard ratio from competing risk Fine–Gray models. 95% CI: 95% confidence interval. NE: not estimable.

**Table 4 cancers-13-01803-t004:** Fine–Gray models for lymphoma-specific survival in stage I EMZL cohort. (N = 7961, including 515 deaths from lymphoma and 1558 deaths from other causes as competing risk).

		Univariable		Multivariable Model 1		Multivariable Model 2	
Variable	Category	SHR (95% CI)	*p*	SHR (95% CI)	*p*	SHR (95% CI)	*p*
**Age**	<60 years	Reference		Reference		Reference	
	≥60 years	4.39 (3.41, 5.64)	<0.001	4.00 (3.10, 5.15)	<0.001	3.99 (3.10, 5.15)	<0.001
**Sex**	Female	Reference		Reference		Reference	
	Male	1.09 (0.91, 1.29)	0.348	1.18 (0.99, 1.42)	0.064	1.19 (0.99, 1.42)	0.059
**Race**	Non-Hispanic white	Reference		Reference		Reference	
	Black	0.88 (0.63, 1.22)	0.434	0.99 (0.72, 1.38)	0.975	1.00 (0.72, 1.40)	0.986
	Other	0.95 (0.73, 1.24)	0.699	1.03 (0.78, 1.35)	0.848	1.04 (0.79, 1.37)	0.795
**Ethnicity**	Non-Hispanic	Reference		Reference		Reference	
	Hispanic	0.98 (0.75, 1.29)	0.897	1.16 (0.88, 1.54)	0.292	1.17 (0.88, 1.55)	0.274
**Primary location**	Gastric	Reference		Reference		Reference	
	Skin	0.36 (0.24, 0.55)	<0.001	0.50 (0.33, 0.77)	0.002	0.52 (0.34, 0.81)	0.004
	Ocular adnexa	0.60 (0.44, 0.82)	0.001	0.75 (0.54, 1.04)	0.086	0.79 (0.56, 1.09)	0.153
	GI non-gastric	1.12 (0.83, 1.51)	0.471	1.18 (0.85, 1.63)	0.328	1.15 (0.83, 1.60)	0.400
	Salivary gland	0.85 (0.61, 1.19)	0.347	1.02 (0.70, 1.48)	0.938	1.05 (0.72, 1.54)	0.788
	Lung	1.48 (1.11, 1.97)	0.007	1.44 (1.05, 1.96)	0.022	1.39 (1.02, 1.91)	0.039
	Breast	1.00 (0.62, 1.62)	0.987	1.14 (0.70, 1.85)	0.601	1.15 (0.71, 1.88)	0.570
	Oral	1.26 (0.79, 2.01)	0.338	1.26 (0.77, 2.06)	0.352	1.27 (0.77, 2.08)	0.343
	Thyroid	0.62 (0.31, 1.26)	0.186	0.73 (0.35, 1.52)	0.405	0.77 (0.37, 1.60)	0.483
	Other	1.54 (1.03, 2.31)	0.037	1.59 (1.05, 2.40)	0.029	1.60 (1.05, 2.44)	0.028
**DLBCL transformation**	No	Reference		Reference		Reference	
	Yes	5.24 (3.74, 7.34)	<0.001	4.63 (3.29, 6.52)	<0.001	4.67 (3.30, 6.60)	<0.001
**Treatment**	RT only	Reference		Reference			
	CT only	2.84 (2.08, 3.87)	<0.001	2.30 (1.68, 3.15)	<0.001		
	SG only	1.42 (1.05, 1.91)	0.021	1.24 (0.90, 1.71)	0.196		
	Combinations	1.20 (0.88, 1.62)	0.246	1.24 (0.90, 1.71)	0.179		
	Observation	1.49 (1.14, 1.95)	0.004	1.29 (0.98, 1.69)	0.067		
**Treatment**	RT only	Reference				Reference	
	CT only	2.84 (2.08, 3.87)	<0.001			2.33 (1.70, 3.19)	<0.001
	SG only	1.42 (1.05, 1.91)	0.021			1.25 (0.90, 1.73)	0.180
	CT + RT	1.33 (0.77, 2.31)	0.305			1.27 (0.74, 2.19)	0.385
	SG + CT	2.33 (1.50, 3.62)	<0.001			1.96 (1.23, 3.11)	0.005
	SG + RT	0.88 (0.60, 1.28)	0.494			0.97 (0.65, 1.44)	0.868
	SG + CT + RT	1.18 (0.52, 2.67)	0.696			1.35 (0.59, 3.07)	0.472
	Observation	1.49 (1.14, 1.95)	0.004			1.30 (0.99, 1.71)	0.058

SHR: subdistribution hazard ratio. 95% CI: 95% confidence interval. Abbreviations: EMZL, extranodal marginal zone lymphoma; GI, gastrointestinal; DLBCL, diffuse large B cell lymphoma; CT, chemotherapy; RT, radiation therapy; SG, surgery.

## Data Availability

Surveillance, Epidemiology, and End Results (SEER) Program. http://www.seer.cancer.gov (accessed on 2 November 2020). SEER*Stat Database: Incidence—SEER 18 Regs Research Data + Hurricane Katrina Impacted Lousiana Cases, November 2018 Sub (1975–2016 varying)—Linked to County attributes—Total U.S., 1969–2017 Counties, National Cancer Institute, DCCPS, Surveillance Research Program.

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
