# Peer review of "Treatments and Outcomes in Stage I Extranodal Marginal Zone Lymphoma in the United States"

_cancers, 2021, doi:10.3390/cancers13081803_

Round 1
Reviewer 1 Report
The study performed by Juan et al. retrospectively analyzed 7961 stage I EMZL patients from the SEER database, and highlighted the therapeutic value of RT in the treatment for early stage EMZL. The study is well carried out with a thorough data analysis. The major concern is the lack of antibiotic treatment for the gastric EMZL. This would compromise the significance of the study, given that stomach is the most frequent involved site and the antibiotics are the most commonly involved treatment. Adding antibiotics to the analytic combination will provide a great value to the study.
Author Response
Reviewer 1
The study performed by Juan et al. retrospectively analyzed 7961 stage I EMZL patients from the SEER database, and highlighted the therapeutic value of RT in the treatment for early stage EMZL. The study is well carried out with a thorough data analysis. The major concern is the lack of antibiotic treatment for the gastric EMZL. This would compromise the significance of the study, given that stomach is the most frequent involved site and the antibiotics are the most commonly involved treatment. Adding antibiotics to the analytic combination will provide a great value to the study.
Response: We agree with the reviewer that data on presence of H, pylori and the use of antibiotic therapy in gastric EMZL would provide a potential explanation on the outcome of gastric MALT in this cohort. Unfortunately, specific treatments including antibiotics and chemotherapy/immunotherapy drugs are not available in the SEER registry. We stated this limitation in last paragraph of page 16 " The present study carries limitations of large epidemiological studies using SEER database such as lack of information on staging modality, indications for therapy, use of antibiotic therapy in patients with gastric EMZL, H. pylori eradication, relapses, progression free survival, radiation doses, chemotherapy regimens and treatment-related adverse events". Despite this, presented analyses provide valuable information on this rare disease and allows to reach conclusions that would not be reached in single or multi-institutional studies in this rare disease.
Reviewer 2 Report
The authors present a SEER database analysis of treatment and outcome in stage IE MALT lymphoma of different sites. While the number of patients is high and the methodology is extensive including interesting comparisons to a general US population, the study has limitations that mostly derive from limited clinical data available in the corresponding database. Given the lack of some information such as staging modality, treatment type / indication for systemic therapy and Helicobacter pylori status for gastric patients, I would suggest to rededefine the results / conclusions of this manuscript.
Major comments
- I would recommend focusing on the excellent outcome of radiotherapy, rather than on the comparison of treatment modalities. It appears difficult in the context of such an analysis to draw conclusions about the superiority of one treatment type over the other. It would be helpful to highlight that in view of the fact that there is no data on treatment indication (an important factor for treatment decision in patients receiving systemic therapy), systemic treatment type and staging modalities applied (advanced dissemination status is often only detected by extensive staging measurements including endoscopy and further site specific imaging), it is no feasible to strongly recommend radiotherapy as the preferred treatment for all stage IE patients. Also we are missing on the Helicobacter / eradication status in gastric patients. I would suggest to concentrate on the general message that patients receiving radiotherapy for stage I disease face a good prognosis even if compared to a normal healthy population.
- Combined modality treatment is rather unusual in MALT lymphoma and not part of current treatment concepts in guidelines, thus conclusions regarding the benefit of combination therapy should be stated with caution. I suggest adding this to the discussion. I would also ask to withdraw this finding from the abstract. Do you have any explanation for the high number of patients receiving combined modality treatment?
- In view of this, I also do not really see the benefit of the second multivariate analysis and further splitting the combined modality treatment in subgroups for multivariate analysis (Table 2). Personally, I think that a descriptive analysis of this is sufficient.
- The missing data on Helicobacter pylori status and eradication is a limitation that cannot be changed; however, I would suggest extending a little bit more on this finding in the discussion. How many patients in the observation group had gastric MALT lymphoma and do I understand correctly that patients receiving eradication were part of this group?
- Another limitation is that the primary stage of disease may depend on the staging modality applied (see also 1.), please add this to the discussion.
Minor comments
- Abstract: Up to one third of patients present with disseminated disease if extensive staging procedures are applied. I would suggest relativizing the first sentence to something like “a considerable amount of patients presents with limited stage…
- Figure 1: Please explain the cohort “ no first malignancy”
Author Response
Reviewer 2
The authors present a SEER database analysis of treatment and outcome in stage IE MALT lymphoma of different sites. While the number of patients is high and the methodology is extensive including interesting comparisons to a general US population, the study has limitations that mostly derive from limited clinical data available in the corresponding database. Given the lack of some information such as staging modality, treatment type / indication for systemic therapy and Helicobacter pylori status for gastric patients, I would suggest to redefine the results / conclusions of this manuscript.
Response: We agree with the reviewer about these limitations of SEER database and have included statement on limitations of this study (last paragraph in page 16). However, SEER database allows the analysis a large number of patients with a rare disease such as stage I EMZL where prospective large scale studies are unlikely to be carried out.
Revised paragraph in page 16 now reads "The present study carries limitations of large epidemiological studies using SEER database such as lack of information on staging modality, indications for therapy, use of antibiotic therapy in patients with gastric EMZL, relapses, progression free survival, radiation doses, chemotherapy regimens and treatment-related adverse events. Clinical and laboratory variables are not available in SEER database preventing the confirmation of previously described predictors of increased risk of transformation. Despite these limitations, SEER database allows statistically powered analysis of a large numbers of patients with a rare and heterogeneous disease such as EMZL".
Major comments
- I would recommend focusing on the excellent outcome of radiotherapy, rather than on the comparison of treatment modalities. It appears difficult in the context of such an analysis to draw conclusions about the superiority of one treatment type over the other. It would be helpful to highlight that in view of the fact that there is no data on treatment indication (an important factor for treatment decision in patients receiving systemic therapy), systemic treatment type and staging modalities applied (advanced dissemination status is often only detected by extensive staging measurements including endoscopy and further site specific imaging), it is no feasible to strongly recommend radiotherapy as the preferred treatment for all stage IE patients. Also we are missing on the Helicobacter / eradication status in gastric patients. I would suggest to concentrate on the general message that patients receiving radiotherapy for stage I disease face a good prognosis even if compared to a normal healthy population.
Response: We thank the reviewer for this important question. We agree that knowledge of reasons triggering systemic therapy initiation in an indolent disease such as EMZL might enlighten us further why these treatments were selected; however, t this data is not available in the SEER. Nevertheless, SEER database captures treatment decisions from a large variety of physicians across the United States and provides important data on the current approach of this disease. Furthermore, as long as the disease is stage I there is indication to treat these patients with curative attempt and our analysis confirms that for the first time, since outcome of stage I patients treated with RT was not different from general population. In the US LymphoCare study of stage I follicular lymphoma, in which indications for treatment were known, only 22% of US patients were treated with radiation, despite potential curative value of this approach (Friedberg et al. JCO 2012, 30:3368). The same was shown in the National Cancer Base Analysis on early stage FL (Vargo et al. Cancer 2015, 121:3325). In that study that also included patients with stage II disease, factors associated with no use of radiation included : increasing age, female sex, African American race, increase comorbidity score, treatment at an academic/research program, Stage II disease, presence of B symptoms and more recent years of diagnosis. Similarly, as stated in the discussion, Olszewski et al. identified factors associated with lower use of RT in patients with stage I/II EMZL including age older than 70 years (OR=0.70, 95%CI 0.59-0.83; P=0.0001), Hispanic (OR= 0.67, 95%CI 0.50-0.89; P=0.006), Black ethnicity (OR=0.48, 95%CI 0.33-0.69; P=0.0001), non-Asians women (OR=0.83, 95%CI 0.73-0.94; P=0.005), areas with higher poverty levels (OR=0.71, 95%CI 0.60-0.85; P=0.0002), rural counties (OR=0.52, 95%CI 0.31-0.87; P=0.01), uninsured patients (OR=0.45, 95%CI 0.22-0.92; P=0.03) and state-sponsored Medicaid insurance (OR=0.55, 95%CI 0.38-0.80; P=0.002) [23] These findings suggest that it is not an indication for treatment but fear to use radiation by US oncologists and other factors as stated above that drive decisions on initial treatment and use of non-Radiation treatments. We have added this to revised discussion. Furthermore, there is no agreement on extent of staging in MZL patients, and while some authors recommend routine neck US and endoscopy in every stage I patient, irrespective of original site, others do not. For example, our group previously demonstrated the absence of prognostic value of staging bone marrow biopsy in patients with clinical/radiological stage I EMZL treated with frontline radiation therapy (Alderuccio JP Blood 2020 PMID: 31978219). In that study, staging bone marrow biopsy did not affect lymphoma-specific survival in this population. Therefore, we do not think that knowledge of the indication for treatment would significantly change the data presented. However, we agree with the reviewer that radiation therapy may not be feasible in some anatomic locations such as GI-nongastric, kidney, and liver, and the use of this approach in patients with salivary gland EMZL, especially associated with Sjögren's, is debatable due to long-term xerostomia and high-risk for subsequent contralateral EMZL. We also agree with the reviewer about the importance to confirm H pylori eradication in gastric EMZL but unfortunately this data is not available in the SEER database and mentioned as a limitation of usage of this data set. .
We would also like to remark that all patients with stage I EMZL in the SEER database experienced shorter overall survival compared to US-matched population; however, when we compared patients treated with radiation therapy survival difference is no longer observed.
Revised discussion now reads (first paragraph in page 16) "Based on these real world results, RT-based therapy should remain the frontline therapy in stage I EMZL. We acknowledge this approach may not be feasible in some anatomic locations such as GI-nongastric or kidney, or long-term toxicity may outweigh benefits like in patients with salivary gland involvement. Further, previous reports on different outcomes in patients with stage I EMZL might result from differences in therapeutic approaches across the studies".
- Combined modality treatment is rather unusual in MALT lymphoma and not part of current treatment concepts in guidelines, thus conclusions regarding the benefit of combination therapy should be stated with caution. I suggest adding this to the discussion. I would also ask to withdraw this finding from the abstract. Do you have any explanation for the high number of patients receiving combined modality treatment?
Response: We agree with the reviewer. We do not have an explanation why this number of patients received combined treatment modality. However, not every patient in large countries like US is treated by an expert in the field. In the LymphoCare study of stage I follicular lymphoma, 22% of patients also received combination of rituximab with chemotherapy as the first line of treatment (Friedberg et al. JCO 2012, 30:3368). Some are treated by general oncologists and SEER database allows us to see what is happening in the real world and perhaps discover unexpected results that can help make conclusions that should prevent future use of inappropriate treatments. For completeness of the data we performed the shown analyses; however, our main conclusion and take-home message is that Radiation therapy is the best approach. In the revised manuscript we reemphasize this in the discussion
- In view of this, I also do not really see the benefit of the second multivariate analysis and further splitting the combined modality treatment in subgroups for multivariate analysis (Table 2). Personally, I think that a descriptive analysis of this is sufficient.
Response: We thank the review for this comment and multivariable analysis model 2 was deleted in Table 2.
- The missing data on Helicobacter pylori status and eradication is a limitation that cannot be changed; however, I would suggest extending a little bit more on this finding in the discussion. How many patients in the observation group had gastric MALT lymphoma and do I understand correctly that patients receiving eradication were part of this group?
Response: This is an important question and reviewer's interpretation is correct. Patients treated with antibiotic therapy were included in observation group. A total of 3031 patients with gastric EMZL were included. Of those patients, 1543 were placed on observation (supplemental Figure 1A). Revised Discussion (last paragraph in page 16) now reads:
" In our study, 3031 patients with gastric EMZL were included. Of those patients, no therapy was recorded in the SEER database for 1543 (50.9%) patients. SEER database does not record presence of H. pylori infection and H. pylori antibiotic therapy, but significant proportion of these patients might be classified under the observation group. However, the exact number of these patients cannot be estimated form the SEER database. We also cannot assume that all these patients were treated with antibiotics, since survival of gastric EMZL stage I patients in the observation group was inferior compared to patients treated with RT. Absence of this data represents one of the major caveats in this analysis. If this data would be available, we might better understand the findings of inferior outcome of gastric EMZL in the SEER database”.
- Another limitation is that the primary stage of disease may depend on the staging modality applied (see also 1.), please add this to the discussion.
Response: Staging modality limitations has been added in the discussion. Revised Discussion now reads (second paragraph page 17) "Extensive staging work-up was postulated by Raderer et al. in EMZL. Staging consisted of ophthalmologic examination, ear, nose and throat investigation including sonography or magnetic resonance imaging of the salivary glands and lacrimal glands, endoscopies with multiple biopsies of the GI tract, computed tomography of thorax and abdomen, sonography of cervical, inguinal, and axillary lymph nodes, and bone marrow biopsy. Twenty-five percent of patients with gastric EMZL had multiorgan involvement beyond the GI tract and 46% of patients with extragastric EMZL also presented advanced disease. Survival was not affected by extension of disease and this approach has not been generalized [39]. However, our group demonstrated the absence of prognostic value of staging bone marrow biopsy in patients with clinical/radiological stage I EMZL treated with frontline RT [40]. In this study staging bone marrow biopsy did not affect lymphoma-specific survival providing further rationale against extensive work up in EMZL".
Minor comments
- Abstract: Up to one third of patients present with disseminated disease if extensive staging procedures are applied. I would suggest relativizing the first sentence to something like “a considerable amount of patients presents with limited stage…
Response: First sentence of the abstract has been edited as requested by the reviewer. Edited abstract now reads: "A considerable number of patients with extranodal marginal zone lymphoma (EMZL) are diagnosed with stage I disease".
- Figure 1: Please explain the cohort “ no first malignancy”
Response: We modified the flowchart as “first primary malignancy” to match the original SEER description. The SEER registries collect the number (but not the site or behavior) of cancers that occur prior to the start of the registry, or prior to the person moving to a SEER catchment area. SEER make the assumption that these cancers are malignant, as is true of the majority of SEER cancers. Thus, if SEER indicates that the first SEER registered tumor is the person's second tumor (the other was a non-SEER cancer) this person's cancers are not considered “first primary malignancy”. Cause of specific death (cancer vs non-cancer related) that is used for competing risk analysis is only limited to first primary malignancies.
Reviewer 3 Report
This is a well-conducted study with a great number of patients for a rare disease (specific histology/localized stage). It provides a thorough examination of site-specific disease where uniformity of treatment is generally lacking.
As correctly acknowledged by the Authors, a great limitation of the study is the lack of a PFS evaluation and analysis, which is unavailable from registry interrogation. However, PFS is one of the most relevant outcomes to be evaluated in MALT lymphomas, where OS is generally unaffected.
Authors refer generically to CHEMOTHERAPY when they consider a systemic treatment. Single agent rituximab or, alternatively, chemoimmunotherapy, is much more appropriate as a systemic therapy in localized MALT, and many institutions rely on rituximab as a pivotal component of the treatment. Is it possible to extrapolate data on the use of rituximab (either as single agent or as chemoimmunotherapy)? This information will improve this work greatly.
Author Response
Reviewer 3
This is a well-conducted study with a great number of patients for a rare disease (specific histology/localized stage). It provides a thorough examination of site-specific disease where uniformity of treatment is generally lacking.
As correctly acknowledged by the Authors, a great limitation of the study is the lack of a PFS evaluation and analysis, which is unavailable from registry interrogation. However, PFS is one of the most relevant outcomes to be evaluated in MALT lymphomas, where OS is generally unaffected.
Authors refer generically to CHEMOTHERAPY when they consider a systemic treatment. Single agent rituximab or, alternatively, chemoimmunotherapy, is much more appropriate as a systemic therapy in localized MALT, and many institutions rely on rituximab as a pivotal component of the treatment. Is it possible to extrapolate data on the use of rituximab (either as single agent or as chemoimmunotherapy)? This information will improve this work greatly.
Response: We agree with the reviewer about importance to assess progression-free survival in low-grade lymphomas. As pointed out by the reviewer SEER database do not collect data on relapse, thus, analysis on progression-free survival is not possible in our study.
We also agree about the importance to know what specific chemotherapy drugs patients received. Particularly in EMZL where single agent rituximab is and accepted frontline therapy. In the SEER database, rituximab is considered chemotherapy for diagnosis years 1999 to 2012 and it is excluded from chemotherapy variable after 2012. However, rituximab administration cannot be determined in SEER. The treatment administered is the captured treatment not only at diagnosis but throughout the course of EMZL. When chemotherapy is identified as “yes”, SEER does not separate rituximab from conventional chemotherapy or number of therapy lines.
This information is now included in the revised manuscript (first paragraph on page 3) and now reads "In the SEER database, rituximab is considered chemotherapy for lymphoma diagnosed between 1999 to 2012 and it is excluded from chemotherapy variable after 2012. The treatment given is the captured treatment not only at diagnosis but throughout the course of EMZL. When chemotherapy is identified as “yes”, SEER does not separate rituximab from conventional chemotherapy or number of therapy lines. This study was conducted according to the guidelines of the Declaration of Helsinki and because SEER data is publically available no approval was requested from the University of Miami IRB or Ethical Committee".
Round 2
Reviewer 2 Report
All comments have been adequately addressed and incorporated in the manuscript.